# Finite temperature effects on Majorana bound states in chiral $p$-wave superconductors

Henrik Schou Røising[1★], Roni Ilan[2], Tobias Meng[3], Steven H. Simon[1] and Felix Flicker[1†]

**1** Rudolf Peierls Center for Theoretical Physics, Oxford OX1 3PU, United Kingdom
**2** Raymond and Beverly Sackler School of Physics and Astronomy,
Tel-Aviv University, Tel-Aviv 69978, Israel
**3** Institut für Theoretische Physik, Dresden 01069, Germany

★ henrik.roising@physics.ox.ac.uk
† flicker@physics.org

## Abstract

We study Majorana modes bound to vortex cores in a chiral $p$-wave superconductor at temperatures non-negligible compared to the superconducting gap. Thermal occupation of Caroli de Gennes-Matricon (CdGM) states, below the full gap, causes the free energy difference between the two fermionic parity sectors to decay algebraically with increasing temperature. The power law acquires an additional factor of $T^{-1}$ for each bound state thermally excited. The zero-temperature result is exponentially recovered well below the minigap (lowest-lying CdGM level). Our results suggest that temperatures larger than the minigap may not be disastrous for topological quantum computation. We discuss the prospect of precision measurements of pinning forces on vortices as a readout scheme for Majorana qubits.


# 1   Introduction

In topological superconductors Majorana modes appear as neutral zero energy excitations associated with defects in the superfluid-like domain walls and vortex cores [1–5]. Majorana modes are candidate non-Abelian anyons [6–8], with direct applications to topological quantum computation [9], and so these collective modes have stimulated intense fundamental and applied research [10, 11].

Candidates for realizing Majorana modes have included nanowire systems [12–18], proximity-induced superconductivity on topological insulators or quantum anomalous Hall insulators [3, 19–24] and unconventional iron based superconductors [25–27] like $(Li_{1-x}Fe_x)OHFeSe$ [28, 29]. Interferometry protocols have been proposed to verify the existence of Majorana modes [4, 5, 30, 31], often subject to restrictions imposed by the (generally low) bulk quality of topological insulators [32, 33].

In a $p$-wave superconductor, two vortex bound states of finite separation $R$ hybridize and split in energy [34]. The splitting effectively produces a finite-energy two-level system of the Majorana qubit, with the levels distinguished by the fermion parity: two Majoranas can 'fuse' into a state with either zero or one fermions (two states of different parity), like two spin-1/2 particles can combine into either a spin-0 or spin-1 state. In some cases, a barrier to exploiting the vortex bound states in topological superconductors is the mixing with the tower of excited states in the vortex cores, known as the Caroli de Gennes-Matricon (CdGM) states [35–37]. The CdGM states are characterized by a level spacing, the 'minigap', of size $\delta_\varepsilon \approx \Delta_0^2/E_F$, with $\Delta_0$ the full superconducting gap and $E_F$ the Fermi energy. Exciting these CdGM states does not lead to loss of quantum information, but it can hide the information so that it is very difficult to manipulate or to measure [38, 39] .

Consider a qubit made from two vortices each having one Majorana zero mode. To measure the state of this qubit, one might, in principle, measure the force between the vortices while moving them close together. At zero temperature there would be a difference in forces for the two different qubit states. At finite temperatures, excitation of the CdGM states reduces the force difference between the two different qubit states, although the quantum information is not lost until temperatures high enough that a bulk quasiparticle is excited that can carry away the fermionic parity [38]. It is thus generally assumed that temperature should be minimized as far as possible in realistic Majorana setups.

In this paper we examine the impact of thermal occupation of the CdGM states on the Majorana energy splitting in a two-vortex system. In a simple analytical model we find that the free energy difference between fermionic parity sectors lies exponentially close to the zero-temperature result for temperatures below the minigap, and decreases only as $T^{-2}$ with increasing temperature $T$ just above this temperature (and well below the superconducting gap).

More generally, the contrast in free energy between parity sectors as two Majoranas are fused decreases with an additional factor of $T^{-1}$ for each thermally-occupied bound state. The polynomial dependence suggests that temperature need not be an immediately limiting factor for topological quantum computation. This result relies, however, on having a 'sizable' minigap. If the vortex core is swamped with a continuum of in-gap states, we find the parity contrast to be exponentially suppressed in temperature. We discuss experimental aspects, including limiting time scales from quasiparticle poisoning and thermal vortex motion, towards the end of the paper.

## 2 Background: Majorana bound states in the $p + ip$ model

We consider an effective spinless $p_x + ip_y$ superconductor in two dimensions, described in the Bogoliubov-de Gennes (BdG) formalism in terms of a coupled eigensystem in the particle-hole basis [40],

$$\mathcal{H}\begin{pmatrix} u_n(\mathbf{r}) \\ v_n(\mathbf{r}) \end{pmatrix} = \varepsilon_n/2 \begin{pmatrix} u_n(\mathbf{r}) \\ v_n(\mathbf{r}) \end{pmatrix}, \tag{1}$$

where $u_n$ ($v_n$) is the particle (hole) component of the eigenstate, and

$$\mathcal{H} = \begin{pmatrix} -\frac{1}{2m}\nabla^2 - E_F & \frac{1}{2k_F}\{\Delta(\mathbf{r}), \partial_x + i\partial_y\} \\ -\frac{1}{2k_F}\{\Delta^*(\mathbf{r}), \partial_x - i\partial_y\} & \frac{1}{2m}\nabla^2 + E_F \end{pmatrix}. \tag{2}$$

Here $\Delta(\mathbf{r})$ is the pairing function at position $\mathbf{r} = (x, y)$, $k_F$ and $E_F$ are the Fermi wavevector and Fermi energy, $m$ is the effective electron mass, and $\varepsilon_n/2$ is the energy of level $n$ of the BdG spectrum. The BdG Hamiltonian in Eq. (2) features a particle-hole-symmetry which ensures that the resulting spectrum is symmetric around zero, such that $\varepsilon_n > 0$ denote the energy differences between the particle and the hole states. Formally, the model belongs to class D in the classification of non-interacting topological superconductors and insulators [41, 42].

The quasiparticle annihilation (creation) operator $\hat{\gamma}_n^{(\dagger)}$ associated with level $\varepsilon_n$ of the BdG Hamiltonian is a superposition of the spinless electron annihilation (creation) operators $\hat{\psi}^{(\dagger)}(\mathbf{r})$,

$$\hat{\gamma}_n = \int d^2 r \left[ u_n^*(\mathbf{r})\hat{\psi}(\mathbf{r}) + v_n^*(\mathbf{r})\hat{\psi}^\dagger(\mathbf{r}) \right]. \tag{3}$$

The operators associated with localized Majorana (zero) modes obey the defining criterion $\hat{\gamma}_n = \hat{\gamma}_n^\dagger$ and the anticommutation relations $\{\hat{\gamma}_n, \hat{\gamma}_m\} = 2\delta_{nm}$.

Vortices in the superconductor are regions around which the complex phase of the order parameter $\Delta$ winds through $2\pi$, with the gap magnitude going to zero in the vortex center. At position $\mathbf{r}$, defining $r = |\mathbf{r}|$ and $\varphi = \arg(\mathbf{r})$, this is described by the order parameter $\Delta(\mathbf{r}) = \Delta_0 e^{i\ell\varphi} f(r)$. Here $\Delta_0$ is the full (asymptotic) gap, $\ell$ the vorticity which is fixed to $\ell = 1$ in the following, and $f(r)$ is the radial profile, where $f(r) \sim r^{|\ell|}$ close to the vortex core. Assuming a profile $f(r) = \tanh(r/\xi)$, where $\xi = v_F/\Delta_0$ is the coherence length and $v_F$ is the Fermi velocity, the Majorana zero mode solution of Eq. (1) for a vortex in an infinite system with $2mE_F > \Delta_0^2/v_F^2$ is explicitly given by [2, 40]

$$\begin{pmatrix} u_0(\mathbf{r}) \\ v_0(\mathbf{r}) \end{pmatrix} = \mathcal{N}\frac{J_1\left(r\sqrt{2mE_F - 1/\xi^2}\right)}{\cosh(r/\xi)} \begin{pmatrix} ie^{i\varphi} \\ -ie^{-i\varphi} \end{pmatrix}, \tag{4}$$

manifestly satisfying the Majorana condition $u_0 = v_0^*$. Here, $\mathcal{N}$ is a normalization constant, and $J_1$ is a Bessel function of the first kind. With $N$ vortices centred at positions $\mathbf{R}_j$, the order

parameter may be approximated by

$$\Delta(\boldsymbol{r}) = \Delta_0 \prod_{j=1}^{N} f(|\boldsymbol{r} - \boldsymbol{R}_j|) e^{i \arg(\boldsymbol{r} - \boldsymbol{R}_j)}, \tag{5}$$

when assuming no spatial or phase fluctuations of the vortices, limiting the validity to the (zero temperature) mean field result for the gap profile [34].

Within the ground state manifold defined by a collection of Majorana zero modes, the pairwise exchanges of excitations constitute a higher-dimensional representation of the (non-Abelian) braid group generators, which is encoded by the unitary operators $\hat{U}_{n,n+1} = \exp\left(-\frac{\pi}{4}\hat{\gamma}_n\hat{\gamma}_{n+1}\right)$ [10, 43]. The braiding operators do not span all unitary gates needed for universal quantum computation, but the remaining set of gates can be implemented in a non-topological way with arbitrarily small errors [11, 44].

With two vortices of finite separation $R = |\boldsymbol{R}_1 - \boldsymbol{R}_2|$ in a topological superconductor, each containing a Majorana mode, the energy splitting between the Majorana modes is given by [34]:

$$\varepsilon_0 \approx \frac{4\Delta_0}{\pi^{3/2}} \frac{\cos\left(k_F R + \frac{\pi}{4}\right)}{\sqrt{k_F R}} e^{-R/\xi}. \tag{6}$$

This expression holds in the regime $R \gg \xi \gg k_F^{-1}$. The hybridization energy above, $\varepsilon_0 = \varepsilon_+ - \varepsilon_-$, is the energy difference between the two fermionic parity sectors, which, when the vortices are well-separated, are associated with the two-vortex wavefunctions $\boldsymbol{\Psi}_\pm = (\boldsymbol{\Psi}_1 \pm i\boldsymbol{\Psi}_2)/\sqrt{2}$, where $\boldsymbol{\Psi}_n = \left(u_n(\boldsymbol{r}), v_n(\boldsymbol{r})\right)^{\mathsf{T}}$ (with $\mathsf{T}$ denoting the transpose) is the wavefunction associated with vortex $n$.

## 3 Effect of heating on CdGM states

The lowest-energy levels, the CdGM levels, in the vortex core occur symmetrically about the Fermi energy, and their level spacing is given by the minigap, $\delta_\varepsilon$. The presence of such in-gap states, which derive from the finite density of states at the Fermi level in the normal state of the core, can lead to reduced distinguishability of the Majorana parity sectors [34, 35]. In this section we consider the quantitative effect of occupying these states at finite temperature. More details on the CdGM levels can be found in Appendix A.

### 3.1 The partition function

In the superconducting condensate, where the number of particles is not conserved, we employ the grand canonical ensemble in evaluations of thermal expectation values. Since the (total) fermionic parity, *i.e.* the number of fermions modulo 2, *is* conserved, the proper partition function is projected onto the even (+) and odd (−) parity sectors [45–48]:

$$\mathcal{Z}_\pm = \frac{1}{2} \prod_n e^{\beta \varepsilon_n/2} \Big[ \prod_m (1 + e^{-\beta \varepsilon_m}) \pm \prod_l (1 - e^{-\beta \varepsilon_l}) \Big] = \frac{1}{2} \mathcal{Z}_0 \Big[ 1 \pm \prod_m \tanh(\beta \varepsilon_m/2) \Big], \tag{7}$$

where $\mathcal{Z}_0 = \prod_n 2\cosh(\beta \varepsilon_n/2)$ is the partition function without parity restrictions, $\beta = (k_B T)^{-1}$, and all the products run over the positive energy levels in the BdG spectrum. We ignore states with energies far above $k_B T$ as they are suppressed by Boltzmann factors at low temperatures. The free energy in the $\pm$ parity sectors can be found in the usual way:

$$F_\pm = -\beta^{-1} \log \mathcal{Z}_\pm. \tag{8}$$

## 3.2 Low-temperature model

We consider two vortices at finite separation such that the Majorana modes in the vortex cores split according to Eq. (6). The excited CdGM states of the cores will also split in the same fashion (see Sec. 3.4). As a general and effective description at low temperatures, where only the lowest levels in the tower of CdGM states are thermally activated, we assume that the system has six levels: $\pm\varepsilon_n/2$, with $n \in \{0, 1, 2\}$, $\varepsilon_1 \equiv \delta_\varepsilon - w_1$ and $\varepsilon_2 \equiv \delta_\varepsilon + w_2$, where $\delta_\varepsilon$ is the minigap. Based on the splitting of the zero modes (Eq. (6)), confirmed numerically, the deviation of the excited states from the minigap, $w_1$ and $w_2$ as defined above, are expected to decay with the vortex-vortex separation as $w_i \sim \exp(-R/\xi)$ for $i = 1, 2$.

The two parity sectors $P = \pm 1$ are treated separately, with the associated particle configurations being denoted by $n_\pm$. In the BdG-spectrum these configurations are depicted in Fig. 1. Notice that the number of particle excitations (disks above zero energy in the figure) is therefore even or odd in the respective parity channels. We consider the hierarchy of energy

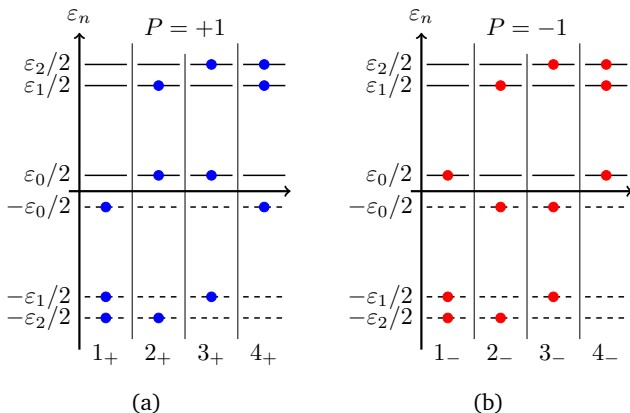

Figure 1: Occupancy configurations, labeled by $n_\pm$ in columns, of the six-level system in the (a) even ($P = +1$) and (b) odd ($P = -1$) parity sector. The single-particle energy levels in the particle-hole symmetric BdG spectrum are denoted by $\pm\varepsilon_n/2$. This depicts a low energy model for a system with finite distance between the two vortices as investigated numerically in Sec. 3.4.

scales $\varepsilon_0 \simeq |w_1| \simeq |w_2| < \delta_\varepsilon$, which we expect to apply when the vortices are further apart than about $3\xi$ (see Sec. 3.4). The partition function is given by

$$\mathcal{Z}_\pm = 2e^{\pm\beta\varepsilon_0/2}\Big[e^{\mp\beta\varepsilon_0}\cosh\big(\beta\frac{w_1+w_2}{2}\big) + \cosh\big(\beta[\delta_\varepsilon + \frac{w_2-w_1}{2}]\big)\Big]. \tag{9}$$

The observable consequence of switching fermionic parity sector is that the free energy involved in bringing two vortices together changes. We term this difference in free energy between the two parity sectors the *parity disparity*,

$$\Delta F \equiv F_- - F_+, \tag{10}$$

which in the current case evaluates to:

$$\Delta F = \varepsilon_0 + \frac{1}{\beta}\log\frac{e^{-\beta\varepsilon_0}\cosh\big(\beta\frac{w_1+w_2}{2}\big) + \cosh\big(\beta[\delta_\varepsilon + \frac{w_2-w_1}{2}]\big)}{e^{+\beta\varepsilon_0}\cosh\big(\beta\frac{w_1+w_2}{2}\big) + \cosh\big(\beta[\delta_\varepsilon + \frac{w_2-w_1}{2}]\big)}. \tag{11}$$

As expected, the parity disparity decreases monotonically with increasing temperature. It reflects the feasibility of directly probing the state of a Majorana qubit. In the limit of low

temperature, $k_B T \lesssim \delta_\varepsilon$, the leading correction to the zero-temperature result is suppressed as $\sim \exp(-\beta \delta_\varepsilon)$:

$$\Delta F \approx \varepsilon_0 - \frac{4}{\beta} \cosh\left(\beta \frac{w_1 + w_2}{2}\right) \sinh(\beta \varepsilon_0) e^{-\beta(\delta_\varepsilon + [w_2 - w_1]/2)}. \tag{12}$$

At temperatures above the minigap we find $F_\pm = -\frac{1}{\beta} \log 4 + \mathcal{O}(\beta \delta_\varepsilon)$ to leading order in the high-temperature expansion, simply reflecting thermal occupation of the four available configurations in each parity sector shown in Fig. 1. The linear corrections in $\beta \delta_\varepsilon$ are independent of parity, such that the parity disparity acquires a leading order correction at second order in $\beta$,

$$\Delta F = \frac{1}{4} \varepsilon_0 \beta^2 \left(\delta_\varepsilon^2 + \delta_\varepsilon(w_2 - w_1) - w_1 w_2\right) + \mathcal{O}(\beta \delta_\varepsilon)^4. \tag{13}$$

This inverse square law decay in the parity disparity with increasing temperature is weak enough that it will likely not preclude direct measurement at temperatures above the minigap. However, this toy model can only describe the regime $k_B T \lesssim 2\delta_\varepsilon$: at higher temperatures further CdGM states will be thermally excited.

Differences between the two parity channels are expected to be washed out at high temperatures. At low temperatures, the fact that the configurations $2_+$ and $3_+$ have excitation energies on the order of $\varepsilon_0$ greater than $2_-$ and $3_-$ (Fig. 1), respectively, distinguishes the two sectors.

## 3.3 Arbitrary numbers of core states

Now consider an arbitrary number of CdGM states thermally active in the two vortex cores. If the vortices are well-separated it is reasonable to assume that $\varepsilon_0 < \varepsilon_1 \approx \varepsilon_2 < \varepsilon_3 \approx \varepsilon_4 < \ldots$, where $\varepsilon_0$ is the Majorana energy, and $\varepsilon_{i>0}$ are the excited CdGM levels. The approximate equalities become exact as $R/\xi \to \infty$. If the temperature is much greater than at least one of the levels, $\varepsilon_n \ll k_B T \ll \Delta_0$ for $n \geqslant 0$, the parity disparity can be approximated by

$$\Delta F = \frac{1}{\beta} \log \frac{1 + \prod_m \tanh(\beta \varepsilon_m/2)}{1 - \prod_n \tanh(\beta \varepsilon_n/2)} \approx \varepsilon_0 (\beta/2)^n \prod_{m=1}^n \varepsilon_m \prod_{l=n+1}^\infty \tanh(\beta \varepsilon_l/2), \tag{14}$$

since $\tanh(\beta \varepsilon_m/2) \approx \beta \varepsilon_m/2 \ll 1 \ \forall m \leqslant n$. Thus, the temperature decay above level $\varepsilon_n$ is algebraic: $\Delta F \sim T^{-n}$. The above decay suggests that temperatures on the order of the minigap can be tolerated but that exciting additional bound states causes the parity disparity to decrease rapidly.

The derivation holds in the limit of a well-defined and sizable minigap. If instead the vortex core contains a continuum of in-gap levels, the parity disparity of Eq. (14) is recast [16] and approximated [1] in the high-temperature limit as

$$\begin{aligned}
\Delta F &= \frac{1}{\beta} \log \coth \left( \frac{1}{2} \int_{\delta_\varepsilon}^\infty dE \ \rho_0 \log \coth(\beta E/2) \right. \\
&\quad \left. + \frac{1}{4} \log \coth(\beta \delta_\varepsilon/2) + \frac{1}{2} \log \coth(\beta \varepsilon_0/2) \right) \\
&\approx \varepsilon_0 \sqrt{2\rho_0 k_B T} \, e^{-\frac{\pi^2}{4} \rho_0 k_B T} e^{1 + \mathcal{O}(1/(\rho_0 k_B T)^2)}.
\end{aligned} \tag{15}$$

Here $\rho_0 \equiv 1/\delta_\varepsilon$ is the density of states in the vortex core. The first term in the middle row of Eq. (15) appears when replacing the sum of equally spaced levels with an integral using the

---

[1]Using that $F(x) = \int_x^\infty dy \log \coth y = \frac{\pi^2}{24} + \frac{1}{2}\left(\log \tanh x \log(1 + \tanh x) + \text{Li}_2(1 - \tanh x) + \text{Li}_2(-\tanh x)\right)$, where $\text{Li}_2$ is the dilogarithm function. For small $x$ this function behaves as $F(x) = \frac{\pi^2}{8} + x(\log x - 1) + \mathcal{O}(x^3)$.

trapezoidal rule. The exponential suppression in $k_B T$ implies that vortex cores with densely packed in-gap states, as on topological insulator surfaces [20] or in superconducting Pb monolayers [49], would likely remain unworkable in the context of readout schemes and topological quantum computation.

## 3.4 Numerical results

Consider two vortices with separation $R$ in a spinless $p + ip$ superconductor. We first assume that $T = 0$ and that the two vortices have the same (positive) phase winding, as in Eq. (5). It should be noted that this is formally different from the vortex-antivortex and the antivortex-antivortex configuration in a $p + ip$ superconductor.

We approach the problem numerically by solving the BdG equations with a finite element method for a large range of inter-vortex distances on a slab no smaller than $[-10\xi, 10\xi] \times [-9\xi, 9\xi]$ with Dirichlet boundary conditions. In Appendix B the calculation is repeated with one vortex replaced by a hole, with a flux quantum penetrating the hole. We fix the (dimensionless) parameters of the $p + ip$ model to $E_F = 3$, $\Delta_0 = 1$, and $mk_F^2 = 54$. The lowest-lying vortex states at zero temperature are shown in Fig. 2. Real-space colour maps of $|\Psi_n(r)| = (|u_n(r)|^2 + |v_n(r)|^2)^{1/2}$ are shown in the figure insets. By following Ref. [35] the first predicted CdGM levels when using the approximation discussed in Appendix A are $\varepsilon_1/2 \approx 0.142\Delta_0$, $\varepsilon_2/2 \approx 0.284\Delta_0$, $\varepsilon_3/2 \approx 0.426\Delta_0$, which are indicated with gray dotted lines in Fig. 2 and agree reasonably well with the numerical results, despite not strictly being in the BCS regime ($\Delta_0 \ll E_F$) where the approximation is valid.

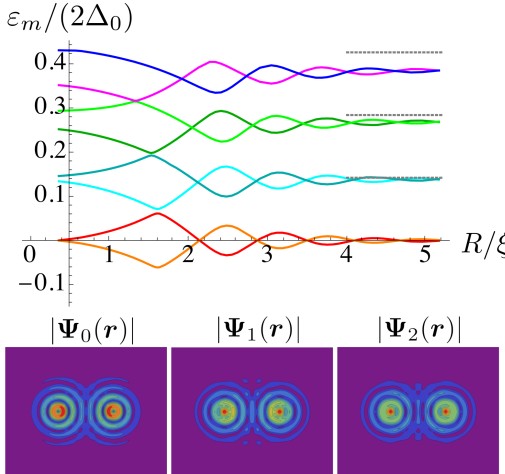

Figure 2: The first energy levels $\varepsilon_m/2$ normalized by the full gap $\Delta_0$ of a two-vortex system. The CdGM levels as predicted with Eq. (19) are shown in gray dotted lines. Colour maps of the core-localized wavefunctions $|\Psi(r)| = (|u_n(r)|^2 + |v_n(r)|^2)^{1/2}$ for the three lowest-lying (positive energy) states are also shown. The wavefunctions are displayed for a vortex-vortex separation of $R/\xi = 3.0$ on a slab of size $8\xi \times 6\xi$.

The CdGM wavefunctions are exponentially localized (with decay length $\xi$) around the two vortex cores, with spatial oscillations set by $2\pi k_F^{-1}$. Close to the vortex cores the expected small argument behaviour of the appropriate Bessel functions (*e.g.* Eq. (4)) is recovered [36]. In the well-separated limit the two vortex cores each host a single Majorana zero mode, and the CdGM levels become doubly degenerate as reflected in Fig. 2. Using the Majorana basis [40] for intermediate separations reveals that the wavefunctions have started to disunite into two localized states at separations as small as about $R/\xi \approx 3$. At larger separations the states $\Psi_\pm$, corresponding to opposite fermionic parity, are reasonable approximations to the true ground

state.

The near alignment of the crossing of the energy levels seen in Fig. 2 is linked to the radial profiles of the vortex states which are given by Bessel functions of the first kind [36], with an argument that increases slightly as a function of the excitation number. The Bessel-type wavefunctions enforce the energy splittings, which derive from the overlaps between the respective states [40], to acquire similar oscillations.

We note that on a finite slab with Dirichlet boundary conditions, the net angular momentum induced by the two vortices enforces boundary states peaked around the edges of the sample, with an energy spacing set by the boundary length, $v_F/L$. When the vortices are located far from the edge compared to $\xi$, these states are not affected by the presence of the vortices or their separation (see Appendix C). Considering instead periodic boundary conditions in both directions (solving the system on a torus) [1] and replacing one vortex by an antivortex [50], removes the boundary states completely.

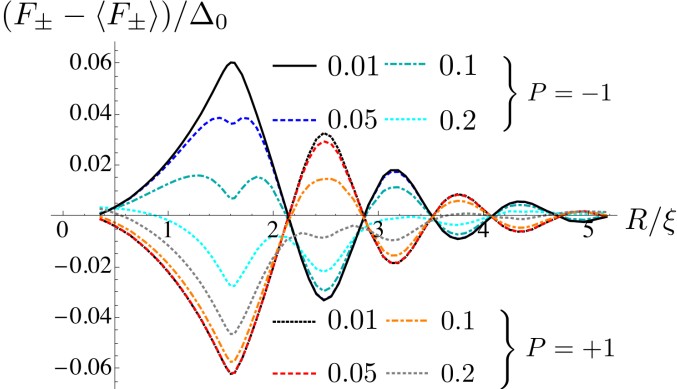

Figure 3: The free energy $F_\pm$ (Eq. (8)) normalized by the full gap $\Delta_0$ in the two parity sectors $P = \pm 1$ as a function of the inter-vortex separation $R/\xi$ for a range of temperatures, with labels referring to the values of $k_B T/\Delta_0$. Here $\langle F_\pm \rangle$ is the free energy averaged over $R/\xi$.

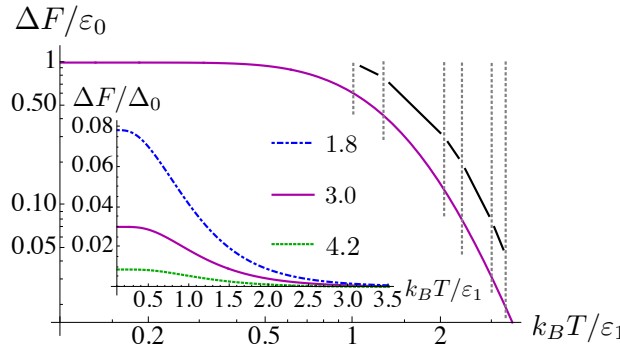

Figure 4: The parity disparity $\Delta F = F_- - F_+$ normalized by the ground state energy level $\varepsilon_0$ as a function of temperature normalized by the excited energy level $\varepsilon_1$. In the main figure (log-log axes) we show the parity disparity for a vortex-vortex separation of $R/\xi = 3.0$. The CdGM levels are indicated with gray dotted lines, and the algebraic temperature law $T^{-n}$, predicted in Eq. (14), is shown in black. The inset shows the parity disparity for three values of the inter-vortex separation, with values of $R/\xi$ given by the labels.

In Figs. 3 and 4 we display the influence of temperature in the presence of in-gap states

for the two-vortex system, when Eq. (8) is applied to the numerically found energy levels of Fig. 2. As the temperature surpasses the minigap, the oscillations begin to smear. Fig. 4 displays the numerically exact temperature dependence of the parity disparity at $R/\xi = 3.0$ in the main figure. The black straight lines represent the simple power law derived below Eq. (14), when replacing $\tanh(\beta \varepsilon_n/2)$ by $\beta \varepsilon_n/2$ for $k_B T > \varepsilon_n$ and 1 for $k_B T < \varepsilon_n$. Measuring the fusion channel of the corresponding qubit as defined by the two Majoranas, and hence the effect of braiding, thus becomes correspondingly difficult at temperatures well above the minigap.

Appendix B contains the same finite-temperature calculation with one vortex replaced by a hole. Qualitatively, the results are very similar to the vortex-vortex case. However, when one vortex is replaced by a hole, and reflection symmetry about $(\boldsymbol{R}_1 + \boldsymbol{R}_2)/2$ in the order parameter is lost, the previous double degeneracy of the levels in the $R/\xi \to \infty$ limit is lifted. The level spacing of the hole states is reduced with the hole circumference (*cf.* Ref. [51]), generally making the issue of a small minigap worse when the hole radius is larger than $\xi$. We note also that the frequency of the CdGM level oscillations with the vortex-hole separation roughly doubles when comparing to the vortex-vortex case.

## 4 Measurement considerations

In practice, the topological splitting of the Majoranas constitutes a small contribution compared to the vortex-vortex repulsion (see Appendix C for a listing of the expressions). The topological contribution will in principle remain clear under the Fourier transform of the free energy. The (Friedel-like) oscillations of frequency $k_F$ from the topological splitting manifest as a bump, with distinguishable features in the two parity channels, revealed for instance in $\mathrm{Re}\{\tilde{F}_- - \tilde{F}_+\}$, with

$$\tilde{F}_\pm(k) \equiv \frac{1}{\sqrt{2\pi}} \int_0^\infty \mathrm{d}R \; e^{-ikR} F_\pm(R),$$ (16)

being the spatial Fourier transform of $F_\pm$. We show this in Fig. 5.

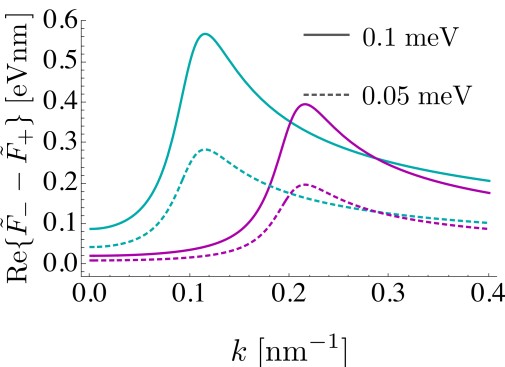

Figure 5: The real part of the Fourier transform $\tilde{F}_\pm$ (expressions listed in Appendix C) of the parity disparity $F_- - F_+$ at zero temperature as a function of momentum $k$. Here, we set the coherence length to the value for Niobium [52], $\xi = 38\,\mathrm{nm}$, and the Fermi momentum to $k_F = 0.1,\ 0.2\,\mathrm{nm}^{-1}$ in cyan and magenta, respectively. The labels refer to the prefactor of the topological contribution in Eq. (6), *i.e.* $4\Delta_0 \pi^{-3/2}$.

By subtracting the free energy before and after a parity flip one is left with a finite result only if the vortices contain non-Abelian anyons. Although the $k_F$ bump in the energy difference would constitute strong evidence of successful braiding, the subtraction of the two curves will

require a sensitivity set by $\varepsilon_0$ in the measured free energy. The measurement series of $F_- - F_+$ can in principle be derived from the corresponding pinning force when one vortex is brought close to another, as measured before ($F_\pm$) and after ($F_\mp$) braiding with a third vortex.

In the following we consider the putative topological superconductor $(Li_{1-x}Fe_x)OHFeSe$ to evaluate the scope of applicability for force measurements to readout of Majorana qubits [28]. The material has a small Fermi energy $E_F \simeq 50 - 60$ meV and a superconducting gap on the order of $2\Delta_0 \simeq 20$ meV. Tunneling measurements on superconducting vortices identify a zero-bias peak, attributed to a Majorana zero mode, clearly separated from excited CdGM levels, with the unusually large minigap $\delta_\varepsilon \simeq 1$ meV $\approx 11$ K.

Numerically, we find the parity disparity at temperatures below the minigap to be roughly $\varepsilon_0 \approx 0.04\Delta_0$ at $R/\xi = 3$. To measure the corresponding pinning force difference between the parity sectors would require a force sensitivity of $\delta F \lesssim \varepsilon_0/\xi \approx 0.05$ pN, with $\xi \simeq 1.4$ nm [28]. This precision is at least an order of magnitude below previously reported force measurement thresholds [53–55], although optimizing these measurements was not the primary goal of those studies.

### 4.1 Timescales and vortex motion

One of the leading threats to topological quantum computation comes from free electrons mixing with the Majorana mode, causing an uncontrolled qubit parity flip [11,38]. The timescale associated with this process is termed the 'poisoning time' [56]. The error rate in a strictly two-dimensional geometry $\Gamma$ is set by the energy scale [57]

$$\Gamma \simeq k_B T e^{-\Delta_0/(k_B T)}. \tag{17}$$

Using $\Delta_0 = 10$ meV for $(Li_{1-x}Fe_x)OHFeSe$, and $T = 5$ K, the expected poisoning time is $t_p \simeq \hbar/\Gamma \approx 20$ ms. Traversing a loop of radius $10\xi$, say, would require a vortex speed of $v \gtrsim 5\,\mu m\,s^{-1}$, which is in principle within reach of existing optical techniques [55]. Another source of qubit decoherence at finite temperature, potentially relevant for this setup, is phonon interactions [58].

Thermal fluctuations change the vortex-vortex distance dynamically, causing smearing over an oscillating function (Eq. (6)). This can greatly reduce the contrast between the parity sectors [34] unless the vortex pinning potential changes significantly on the order of the oscillation lengthscale $2\pi k_F^{-1} \simeq 21$ nm. This approach would therefore require the vortices to be artificially pinned in tight potentials [59] on this scale, if the temperature is on the order of the minigap.

Requiring adiabatic motion, to avoid the excitation of quasiparticles, introduces a lower time limit on the braiding operations. However, this timescale is small for the compound under consideration, $t_a \simeq \hbar/\delta_\varepsilon \approx 0.7$ ps. The braiding operations are therefore restricted to timescales $t_a \ll t \ll t_p$. This could be achievable in the near future if improvements continue to be made to individual vortex manipulation [53–55,59] for topological superconductors with sizable minigaps [27,28].

## 5 Conclusions

In this paper we have demonstrated the effect, on the Majorana energy splitting of two bound states, of thermally exciting the CdGM states in vortex cores. The parity disparity (the difference in free energies between the two Majorana parity channels when the Majoranas are brought close together), which reflects the state of a Majorana qubit, lies exponentially close to the zero-temperature result for temperatures well below the minigap $\sim \Delta_0^2/E_F$. Below the

full gap, thermal excitation of higher CdGM states causes the amplitude to decay algebraically in temperature, with an additional factor of $T^{-1}$ for each CdGM state thermally occupied on average. If the in-gap states are densely packed the suppression in temperature becomes exponential.

The relatively weak decay of the parity disparity with increasing temperature means that temperature does not necessarily need to be minimized in experimental scenarios. In fact, the Majorana modes can in principle coexist with excited states without loss of quantum information in any readout scheme based on the total fermion parity [38]. Local heating can potentially be used as another degree of freedom. This could prove useful, for example, if the magnitude of the external magnetic field is limited by other factors (working at a higher temperature can decrease the critical field $H_{c1}$). Another possible application is to intentionally excite the CdGM modes in order to decouple the vortex motion from the state of the Majorana qubit, thus helping to increase decoherence times of the qubit.

As a promising candidate topological superconductor we draw specific attention to the intrinsic type-II superconductor $(Li_{1-x}Fe_x)OHFeSe$, recently suggested as a possible platform for topological quantum computation [28]. With the results presented in the above sections, this material should have an experimentally accessible temperature range in which one can aim at probing the parity disparity. In the near-term, we hope the knowledge that finite temperature effects need not be disastrous for the measurement and control of Majoranas may open other avenues of enquiry.

## Acknowledgments

We thank Jay D. Sau for suggesting the outcome of a small minigap in Eq. (15). H.S.R. acknowledges discussions with J.M. Leinaas, and is grateful to T. Zhang for useful comments. H.S.R. is supported by the Aker Scholarship. T.M. is supported by the Deutsche Forschungsgemeinschaft via the Emmy Noether Programme ME 4844/1-1 and through SFB 1143. S.H.S. is supported by EPSRC grant numbers EP/I031014/1 and EP/N01930X/1. F.F. acknowledges helpful discussions with P.J.W. Moll, S. Speller, and A. Akhmerov, and support from the Astor Junior Research Fellowship of New College, Oxford.

## A  The CdGM states

The low-energy spectrum of the vortex is given by [36]

$$\varepsilon_{n-(\ell+1)/2} = -\left(n - \frac{\ell+1}{2}\right)\delta_\varepsilon, \tag{18}$$

with $n \in \mathbb{Z}$ the angular momentum of the state, $\delta_\varepsilon$ the minigap, and $\ell \in \mathbb{Z}$ the vorticity. The excited core-localized CdGM states thus disperse linearly in momentum in $p$-wave superconductors. For odd vorticity the vortex hosts a zero energy mode of zero angular momentum relative to the condensate. The minigap for a $p$-wave superconductor can be estimated by approximating the vortex as a hard step in the gap function. By continuity of the wavefunction at the step radius one can deduce [35, 36]

$$\delta_\varepsilon \approx \frac{2m\Delta_0^2}{k_F^2} \frac{\int_0^\infty d\rho\, f(\rho)/\rho\, \exp\left(-2\int_0^\rho d\rho'\, f(\rho')\right)}{\int_0^\infty d\rho\, \exp\left(-2\int_0^\rho d\rho'\, f(\rho')\right)}, \tag{19}$$

where $\rho = r/\xi$ is a dimensionless length. Using $f(\rho) = \tanh(\rho)$ for the vortex profile of unit vorticity [60], and assuming $k_F^2 \approx 2mE_F$ (which holds in the BCS regime $\Delta_0 \ll E_F$), the above

formula yields the level spacing

$$\delta_\varepsilon \approx \frac{7\zeta(3)}{\pi^2}\frac{\Delta_0^2}{E_F},\tag{20}$$

with $\zeta$ the Riemann zeta function. The above formulas yield the gray dotted levels indicated in Fig. 2.

## B The vortex-hole system

In this appendix we list numerical results, similar to those shown in Figs. 2 and 3, for a vortex-hole system. We use the order parameter magnitude $|\Delta(\boldsymbol{r})| = \Delta_0 f_1(|\boldsymbol{r}-\boldsymbol{R}_1|)f_2(|\boldsymbol{r}-\boldsymbol{R}_2|)$, where $\Delta_0$ is the full gap, and $\boldsymbol{R}_1$ ($\boldsymbol{R}_2$) is the position of the vortex (hole). For the gap profiles we take $f_1(r) = \tanh(r/\xi)$ and $f_2(r) = \frac{1}{2}\left(1+\tanh\left(\alpha[r^2/\xi^2 - \eta^2]\right)\right)$ (which is chosen for numerical convenience and approximates a Heaviside step function when $\alpha$ is large), with $\eta = 0.6$ and $\alpha = 10$. The vorticity is $+1$ for both the vortex and the hole, and we solve the BdG equations on a finite slab with Dirichlet boundary conditions as described in Sec. 3.4. The parameters of the $p+ip$ model are taken to be the same as in the aforementioned section.

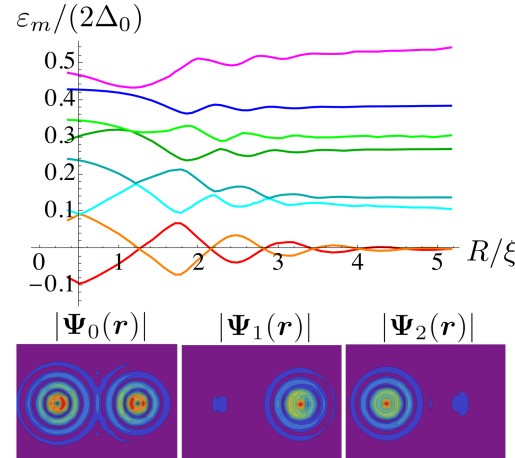

Figure 6: Same as Fig. 2 with one vortex replaced by a hole. In the wavefunction colour maps, shown here for $R/\xi = 5.0$, the (lowest positive energy) state $\boldsymbol{\Psi}_0$ is equally weighted between the vortex and the hole, $\boldsymbol{\Psi}_1$ is mainly localized in the hole (to the right) and $\boldsymbol{\Psi}_2$ is mainly localized in the vortex (to the left).

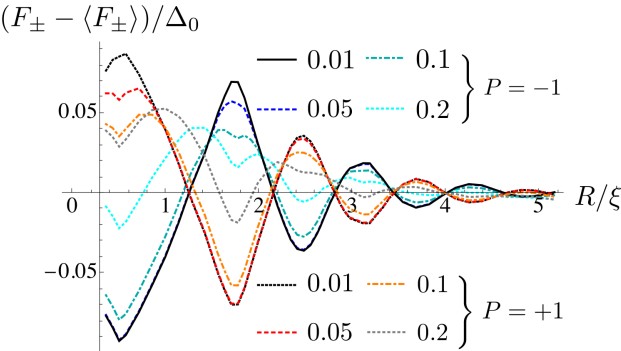

Figure 7: Same as Fig. 3 with one vortex replaced by a hole.

In Fig. 6 the vortex-hole energy levels are shown, along with colour maps of the three lowest-lying wavefunctions. We note that unlike the vortex-vortex case where the CdGM levels are doubly degenerate in the $R/\xi \to \infty$ limit, the levels corresponding to the vortex and the hole are distinguishable. Accordingly, the wavefunctions of the excited states localize around either the hole (right in the figure) or the vortex (left in the figure). In Fig. 7 we show the impact of finite temperature on the parity disparity in the case of a vortex-hole system. Qualitatively, the temperature smearing here is similar to that of the vortex-vortex system in Fig. 3.

## C  Energy contributions

In this appendix we first list the full expression for the spatial Fourier transform (Eq. (16)) of the topological energy contribution (Eq. (6)), which in Sec. 4 was suggested as a key signature for experimentally detecting the result of braiding. Second, we address the question of how the bulk levels depend on the inter-vortex separation.

### C.1  Fourier transforms of the two-vortex energy contributions

The contributions to the vortex-vortex energy are

$$\varepsilon_{\text{top}}(R) = \varepsilon_{0,\text{top}} \frac{\cos(k_F R + \pi/4)}{\sqrt{k_F R}} e^{-R/\xi}, \tag{21}$$

$$\varepsilon_{\text{cl}}(R) = \varepsilon_{0,\text{cl}} K_0(R/\lambda), \tag{22}$$

where $\varepsilon_{0,\text{top}} = 4\Delta_0 \pi^{-3/2}$ from Eq. (6), $K_0$ is a hyperbolic Bessel function, $\varepsilon_{0,\text{cl}} = \Phi_0^2/(4\pi\lambda)^2$ where $\Phi_0 = h/(2e)$ is the flux quantum, and $\lambda$ is the London penetration depth [60]. Defining the Fourier transform as in Eq. (16) yields for the topological contribution

$$\tilde{\varepsilon}_{\text{top}}(k) = \frac{\varepsilon_{0,\text{top}}}{2} \sqrt{\frac{\xi}{k_F}} \left[ (i\xi k + 1) \cosh\left(\frac{3}{2}\text{arctanh}\left(\frac{k_F \xi}{\xi k - i}\right)\right) + i k_F \xi \sinh\left(\frac{3}{2}\text{arctanh}\left(\frac{k_F \xi}{i - \xi k}\right)\right) + \right.$$

$$\left. + (i - \xi k) \sqrt{1 - \left[\frac{k_F \xi}{\xi k - i}\right]^2} \sinh\left(\frac{1}{2}\text{arctanh}\left(\frac{k_F \xi}{\xi k - i}\right)\right) \right]$$

$$\times \left[ (i\xi k + 1)^{3/2} \left(1 - \left[\frac{k_F \xi}{\xi k - i}\right]^2\right)^{3/4} \right]^{-1}, \tag{23}$$

of which the real part is shown in Fig. 5. From the oscillating part of $\varepsilon_{\text{top}}(R)$ there is a characteristic peak in $\tilde{\varepsilon}_{\text{top}}(k)$ at $k = k_F$ that is enhanced with increasing $\xi$.

### C.2  The background energy

The Hamiltonian associated with the mean field description of Eq. (2) can be written in diagonal form:

$$H = \int \mathrm{d}^2 r \left(\hat{\psi}^\dagger(\mathbf{r}), \hat{\psi}(\mathbf{r})\right) \mathcal{H} \begin{pmatrix} \hat{\psi}(\mathbf{r}) \\ \hat{\psi}^\dagger(\mathbf{r}) \end{pmatrix} = \sum_n \varepsilon_n \hat{\gamma}_n^\dagger \hat{\gamma}_n - \frac{1}{2} \sum_n \varepsilon_n, \tag{24}$$

where $\hat{\psi}^{(\dagger)}(\boldsymbol{r})$ is the annihilation (creation) operator of a spinless electron at position $\boldsymbol{r}$, and the quasiparticle annihilation (creation) operator associated with level $\varepsilon_n$ is $\hat{\gamma}_n^{(\dagger)}$.

The oscillations in the energy of the Majorana modes, as given in Eq. (6), could in principle be threatened by a conspiracy of the bulk energy levels, even at zero temperature (deriving from the last term in Eq. (24)). The sum of bulk energies could *a priori* have an oscillatory dependence on the inter-vortex separation $R$, like the individual Majorana and the CdGM levels, and thereby drown out the topological contribution. On a finite slab with Dirichlet boundary conditions, Fig. 8 shows the result of the energy sum for the 150 lowest energy states, including all the low-lying edge states. The oscillations of the Majorana levels at zero

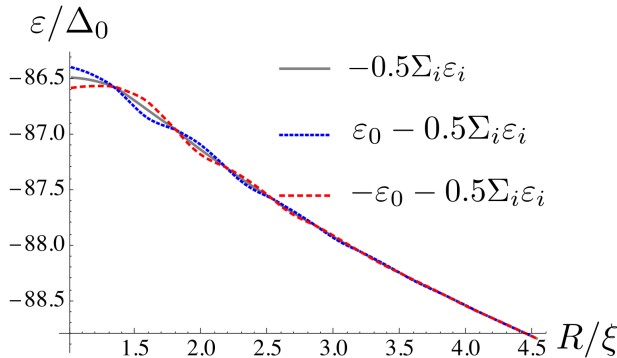

Figure 8: The Majorana energy oscillations as a function of the inter-vortex separation on top of the ground state energy shift (the vortex-vortex repulsion) from Eq. (24) when including the lowest 150 states on a finite slab with Dirichlet boundary conditions.

temperature $\pm \varepsilon_0$ are clearly visible on top of a $\sim \log(R/\xi)$ background that emerges. This background energy we assign to the 'classical' vortex-vortex repulsion that arises from the mutual Lorentz force of the flux lines [60]. We checked that in the case of a vortex-antivortex system on a torus, where there are no edge states, the same background energy emerges with the opposite sign, *i.e.* an attractive contribution.

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
