# Peer review of "Finite temperature effects on Majorana bound states in chiral p-wave superconductors"

_SciPost Physics, doi:SciPost Phys. 6, 055 (2019)_

## Round 1 · Referee Report · Jay Sau (Referee 1) · 2019-3-12

Strengths

(1) I think this is quite an interesting paper because it discusses the read-out of finite
temperature qubits. While Ref 35 showed that the minigap was not a fundamental problem,
it was clear that read-out could be a problem. In fact, the interferometric measurements
considered back then probably suffer from decoherence beyond what is considered in this
work. The discussion force measurements, which are simpler and somewhat more realistic compared to interferometry is very nice.

(2) The central result of the paper i.e. the temperature scaling of the force measurement is likely to provide guidance for how the minigap in vortices affect quantum information applications.

(3) I also think the paper is very well-written so that I don't have any suggestions to improve the presentation.

Weaknesses

(1) While the manuscript is playfully careful to use phrases such as "not disastrous",
I think the summary of Eq. 14 i.e. "decreases with an additional factor of T^-1 for
each thermally-occupied bound state" is not explicit enough about the effect of a small minigap.

(2) In the present manuscript the authors only list inter-vortex force as an incidental example in the introduction as
"If we want to measure the state of this qubit, we might, in principle, try to measure the force between the vortices as we move them close together."
However the analysis of the authors specifically applies to the force approach of measuring Majorana qubits and not others. The present manuscript is not explicit about the scope of applicability of the free-energy as a measurement tool.

(3) The manuscript ignore the thermal motion of the center of the vortex (see Ref 31)

Report

Dear Editor,

The manuscript titled "Finite temperature effects on Majorana
bound states in chiral p-wave superconductors" revisits the role of vortex sub-gap
states for quantum computation using Majorana fermions. This is particularly
relevant for vortex Majorana systems, where the mini-gap might be somewhat small.
The traditional treatment of quantum computation using Majoranas assumes that the
Majorana mode is isolated so that a small minigap is a major obstacle to Majorana
information processing. Following Ref 35, it has been clear that the minigap is not
directly a problem for the storage of quantum information as long as the temperature
is substantially below the bulk gap. The present manuscript is the first one that I am
aware of which explores the read-out of such Majorana qubits in this regime with a
small mini-gap. Quantum computation is not possible without a read-out scheme.
The present manuscript computes the free-energy as a function of distance for a pair
of vortices. As a matter of principle, this free-energy can be measured from the
parity dependence of the force between a pair of vortices. The authors find that a
single minigap state leads to a power-law supression in temperature of the parity.
This result is interpreted as "that temperatures larger than the minigap may not
be disastrous for topological quantum computation."

Requested changes

(1) I think weakness 1 can be remedied by categorically emphasizing the consequence of Eq. 14.
Basically the authors should add an extra expression for the suppression factor to Delta F
which is [Delta F ~ epsilon_0 sqrt(T rho) exp (- T rho)] or something like that,
where rho is the density of states in the vortex (i.e. inverse level spacing).
The main point that needs to be emphasized is that "small" level spacing as would happen
in say an Aluminum superconducting vortex is still bad. I think the authors do a
good job talking about iron superconductors where this would be less of a problem. But
it needs to be explicit that the authors also find that for example Aluminum vortices on
topological insulators will work.

(2) Weakness 2 should be remedied by making the force measurement approach a more central part of the paper by making it clear from the abstract that they consider approaches such as measurement of force (possibly in a generalized sense) that can be used to determine the free-energy. I think this actually enhances the value of the manuscript since the inter-vortex force has not been extensively discussed in the literature as a measurement tool. Alternatively, if the authors got this idea from some source, I think it is important to cite a relevant reference.

(3) With regard to weakness 3, the authors should either mention the thermal vortex motion as an additional factor and preferably estimate its relative effect.

---

## Round 1 · Referee Report · Anonymous (Referee 2) · 2019-3-15

Strengths

1- Finite temperature effects of topological qubits is an important topic for practical realizations. While it was realized earlier (Ref 35) that vortex states above the minigap in principle preserve the quantum information, the paper explicitly shows that parity measurements remain possible at finite temperatures of the order of the minigap.

2- The paper is clearly written, with appropriate additional information in the Appendix.

Weaknesses

1- The connection between the free energy parity disparity that the authors consider and experimentally measurable quantities (the force between vorticies) is not much discussed.

2- The authors write that temperatures above the minigap may not be detrimental. Their results suggest a more specific statement: Temperatures of the order of the minigap can be tolerated but for larger temperatures the parity effect is rapidly decaying.

3- It would be interesting to not only quantify the effect of finite temperatures on the decay of the read out signal but also discuss the effect of dephasing the topological qubit.

Report

In their paper the authors discuss the effect of finite temperature excitations on the read out of topological qubits based on vortex states in p-wave superconductors. This work is particularly timely in light of recent experimental progress consistent with observing Majorana modes in superconducting vortices (Ref 22).

Earlier work (Ref 35) found that, in principle, excited vortex states do not lead to a loss of quantum information. The authors provide an important advance by quantifying the effect of finte-temperature excitations on the read-out visibility of the qubit. To this end, the authors focus on the decay of the free energy difference between the even and odd parity (qubit states). In principle, the free energy difference is measurable via the attraction force between two vortices. A few more details about what forces are experimentally feasible to measure and a comparison to the expected parity effect would improve the paper.

Up to minor revisions I would recommend to accept the paper for publication.

Requested changes

1- Weakness 1 could be addressed by providing references for force measurements between vortices and estimating whether the predicted parity-dependent forces lie in the range (or at least not too far away) of experimental capabilities. Moreover, it would be important to note whether the experiments cited by the authors in Ref 22 have the capability of moving vortices and measuring forces (at least in principle). Also, do vortex pinning centers affect the results of the authors?

2- An explicit statement along the lines above (weakness 2) would be sufficient to address weakness 2.

3- I encourage the authors to consider weakness 3. I am not sure that this can be easily addressed and this might be more appropriate for future work.

---

## Round 2 · Referee Report · Anonymous (Referee 3) · 2019-4-13

Report

With the resubmission I find that the manuscript improved and the authors addressed all my previous comments/questions. However, it seems that the newly introduced discussion of quasiparticle poisoning is actually too pessimistic. The authors estimate the poisoning rate by the thermal activation above the minigap. If Eq. (17) would indeed be the limiting factor then the regime of temperatures of the order of the minigap that is discussed throughout the paper would lead to unpractically short poisoning rates. In fact, Eq. (17) would reintroduce the condition that temperatures much smaller than the minigap are required. Fortunately this is not necessary since the poisoning rate of interest is that of external quasiparticles entering the vortex. Ideally, the latter rate is exponentially suppressed with the full gap of the bulk superconductor. In practice the possible presence of non-equilibrium quasiparticles introduces a more complicated expression for the poisoning rate but this might be difficult to estimate at the current stage.

  • validity: high
  • significance: high
  • originality: good
  • clarity: high
  • formatting: excellent
  • grammar: excellent

Author:  Henrik Røising  on 2019-05-08  [id 508]

(in reply to Report 1 on 2019-04-13)

We thank the referee for their further helpful comments. We agree that, in a strictly two-dimensional geometry, it is the full gap rather than the minigap which should appear in Eq. (17). We have corrected this in the published version of the manuscript.

---

## Round 2 · Author Response

We kindly thank the editor for taking the time to consider our manuscript, and the referees for their useful reports and recommendations. We will address all points raised by the referees. We believe the manuscript is now suitable for publication in SciPost.

Both referees requested that we emphasize the effect of the parity disparity on the inter-vortex force, leading to a protocol for read-out of the qubit state. This is a helpful suggestion, and indeed earlier versions of the draft contained a significantly greater discussion of this point, including detailed calculations for specific experimental set-ups. We found that the required sensitivity of measurement, and precision of control in manipulating the vortices, place force measurements slightly beyond the reach of current techniques, and so we opted to de-emphasise the extent to which we proposed force measurements as a useful protocol. That said, since both referees request further discussion on this point, we have re-introduced a longer discussion of the specific set-up we found to be the best candidate given existing reported data.

Response to referee 1:

We thank Prof. Sau for his positive assessment of our work, and for his insightful and helpful comments and recommendations. We address each of the requested changes in turn.

(1) We have added a new paragraph at the end of Sec. III.C in which we calculate the parity disparity with a continuum of in-gap states. Our result, derived in Eq. (15), matches the expression suggested by Prof. Sau. We emphasize, in the Introduction and Conclusions, the precise manner in which a small level spacing would be detrimental to the read-out of the Majorana qubit parity, providing specific examples.

(2) In order to emphasise possible force measurements we have made Sec. IV a more central part of the paper, extending it with estimates of the required force sensitivity and a discussion of braiding timescales. We now also make this point in the abstract. Several experimental references have also been added to make contact with present technologies. We feel these additions give a fair representation of the plausibility of this approach without over-selling the likelihood of success with present set-ups.

(3) As part of the added subsection IV.A we discuss the consequences of thermal vortex motion and the required properties of the vortex pinning potential to avoid a substantial suppression of the parity disparity.

Response to referee 2:

We again thank the referee for their positive assessment of our work, and their helpful suggestions.

(1) Further to our general comments on force measurements above, we have included specific estimates concerning the compound in Ref. 28 (previously Ref. 22) in the two last paragraphs before subsection IV.A. The suggestion that we discuss the effect of pinning potentials is also helpful. We have added a discussion of pinning potentials in the second paragraph of subsection IV.A.

(2) We agree with the referee that a greater discussion of higher-temperature effects is important, and the discussion combines naturally with the small-minigap effects suggested by the first referee. We have refined the consequences of Eq. (14) by adding a sentence below that equation along the lines suggested by the referee. We have also added another paragraph to Sec. III.C on the consequences of having a continuum of in-gap states. We emphasize these points in the Introduction and Conclusions.

(3) We again thank the referee for this helpful suggestion. We now discuss the poisoning time, leading to dephasing, in subsection IV.A, with associated references added. We agree that quantifying these effects is vital to any realistic proposal for implementing and measuring Majorana braiding; an in-depth study of dephasing effects is outside the scope of the current work, but plays a central role in a follow-up project.

---

## Round 2 · List of Changes

(1) A paragraph (including Eq. (15)) and a sentence have been added to the end of Sec. III.C. The new paragraph addresses the high-temperature limit of the parity disparity with a continuum of vortex in-gap states.

(2) Section IV is extended to include (at the end) two paragraphs on the force measurement consideration in an iron based compound and a new subsection, IV.A, on quasiparticle poisoning and thermal vortex motion in pinning potentials.

(3) Three sentences at the end of the Introduction, and one sentence at the end of the first paragraph of the Conclusions, have been added to emphasize the validity of the main result.

(4) A sentence has been added to the end of the abstract to advertise that we consider applications to read-out of Majorana qubits.

(5) References 9, 17, 20, 25, 27, 49, 53 - 59, and footnote 48 have been added.

---

## Editorial Decision

published